# Effects of Mass Attachments on Flutter Characteristics of Thin-Walled Panels

**Wuchao Qi [1],\* , Meng Wang [1] and Sumei Tian [2]**

1 Key Laboratory of Liaoning Province for Aircraft Composite Structural Analysis and Simulation, Shenyang Aerospace University, Shenyang 110136, China
2 College of Aeroengine, Shenyang Aerospace University, Shenyang 110136, China
\* Correspondence: qiwuchao@sau.edu.cn

**Abstract:** Mass attachments may exist in the design and use of an aircraft panel, such as sensor layout, internal wiring, surface icing, etc. These mass attachments can change the flutter characteristics of the panel in supersonic flight and have important impacts on structural safety. In order to investigate the flutter characteristics of the panel with mass attachments, an assumed mode method is proposed to deal with the changes in the modal properties of the panel structure. Combined with the first order piston theory and *p-k* method, the flutter velocities and flutter frequencies of the panel under different cases can be obtained in the frequency domain. Firstly, based on the large displacement with a small strain assumption proposed by von Kármán and the proposed assumed mode method, the structural dynamic model of a simply supported panel with mass attachments and artificial dampers is constructed. Then, modal aerodynamic forces of the simply supported panel can be obtained based on first-order piston theory. Finally, flutter equations are transformed into the frequency domain and solved by the *p-k* method. The results showed that the existence of mass attachments can significantly change the flutter velocities and flutter frequencies of the panel. However, the flutter characteristics of the panel can be enhanced or recovered through some appropriate damper configuration schemes. Calculating the flutter characteristics of thin-walled panels with mass attachments can more accurately simulate real situations during flight, and one can obtain a safer design scheme of thin-walled panels.

**Keywords:** mass attachment; thin-walled panel; the assumed mode method; flutter characteristic; piston theory





## 1. Introduction

In the process of aircraft structural design, a large number of thin-walled structures are used, such as the skin of wings and fuselages and the rudder surface of missiles. These structures are usually firmly connected to ribs, frames, stringers and other structural members through rivets and other connectors to realize the transmission of external aerodynamic loads. At the same time, they are divided into several small panels that can be approximated as panels simply supported on four sides in a mechanical model. In the process of the aeroelastic design of these panels, the relationships among the mass attributes, stiffness attributes and aerodynamic forces are usually considered, while the mass attachments that may exist in the design and service process are often ignored. However, mass attachments may greatly change the mass distribution of the panel, which in turn affects the flutter characteristics of the panel [1]. For example, engineers often need to arrange some sensors on the thin-walled panels to observe the changes in the stresses and strains of the structure so that they can find and repair damages or cracks in the panel in time. However, these sensors become mass attachments for the panel. In addition, in some emergency situations, it may be necessary to obtain escape routes by blasting some panels. This requires that self-destruction devices are arranged in advance on some panels, which will also bring additional masses to the panels. For another example, there are a large number of wires

inside the aircraft to control the aircraft attitude by transmitting signals. These wires may be attached to a certain position of a panel by bonding, thus bringing mass addition to the panel. Additionally, when the aircraft is flying at a high altitude, the condensation of water vapor in the air may bring additional ice loads to the panel.

In terms of vibration analysis of a panel or a beam with attached masses, Plaut et al. [2] consider a buckled beam with immovable pinned ends that attached concentrated masses in 2016. The effects of various parameters were investigated, such as the ratio of the span to the total arc length of the beam, the locations and weights of the attached masses and systems and the stiffnesses of the springs. Ghadiri et al. [3] proposed an analytical solution method to explore the vibration characteristics of a cantilever functionally graded nanobeam with a concentrated mass exposed to thermal loading for the first time. At the same time, the vibration of symmetrically laminated composite plates with an attached mass was studied by Aydogdu et al. [4]. The Ritz method with algebraic polynomial displacement field was used. Rahmane et al. [5] addressed the effect of an attached mass on the dynamic properties of composite laminate plates, under flexural vibration, for clamped–free–free–free boundary conditions. Avdonin et al. [6] considered the problem of boundary control for a vibrating string with interior point masses. The control problem was reduced to a moment problem, which was then solved using the theory of exponential divided differences in tandem with unique shape and velocity controllability results. Aksencer et al. [7] investigated the free vibration of a rotating laminated composite beam with an attached point mass. The Ritz method with algebraic polynomials was used in the formulation. De Rosa et al. [8] considered the free vibration of a tapered beam modeling nonuniform single-walled carbon nanotubes. The beam was clamped at one end and elastically restrained at the other, where a concentrated mass was also located. Lumentut et al. [9] focused on the primary development of novel analytical and numerical studies for the smart plate structure due to the effects of point mass locations, dynamic motions and network segmentations. Lei et al. [10] employed both the traditional method of the separation of variables and the method of the Laplace transform to solve the eigenvalue problem of the free vibration of such structures. Uymaz [11] analyzed the effects of mass and temperature on the free vibration of the functionally graded plate carrying a point mass at an arbitrary position with a three-dimensional Ritz solution. Kalosha et al. [12] considered a mathematical model of a simply supported Euler–Bernoulli beam with an attached spring–mass system. The model was controlled by distributed piezo actuators and a lumped force. Dadoulis et al. [13] investigated analytical solutions for beams with sizeable mass attachments under externally induced base motions. Mahmoud [14] outlined the use of the Myklestad method (also known as the Lumped Mass Transfer Matrix Method) to conduct a free vibration analysis of nonuniform and stepped axially functionally graded (AFG) beams carrying arbitrary numbers of point masses.

In addition, in terms of beam or panel flutter with mass attachments, Jaiman et al. [15] presented a review and theoretical study of the added mass and aeroelastic instability exhibited by a linear elastic plate immersed in a mean flow. Sohrabian et al. [16] presented the effects of shear deformation on the flutter instability of a cantilever beam subject to a concentrated follower force. Pacheco et al. [17] employed nonlinear energy sinks to suppress panel flutter and reduce the intensity of limit cycle oscillations. Ghasemikaram et al. [18] investigated flutter analysis and the suppression of an aircraft wing with a flexibly mounted external store using a magnetorheological damper. Molina et al. [19] focused on the influence of mass matrix models on the flutter computations of aircraft structures. Bera et al. [20] considered the flutter control of a bridge deck section using a combination of aerodynamic and mechanical measures. Zhou et al. [21,22] used a nonlinear energy sink (NES) to suppress panel flutter. A nonlinear aeroelastic model for a two-dimensional flat panel with an NES in supersonic flow was established using the Galerkin method. The effects of the NES parameters on the flutter boundaries of the panel were investigated using Lyapunov's indirect method. Sun et al. [23] studied the influence mechanism of the lumped

mass on the aeroelastic behaviors of two-dimensional (2D) panels in supersonic airflow, and proposed an axially functionally graded design method using the lumped mass.

In this paper, the structural dynamic model of a panel with mass attachments is constructed based on the proposed assumed mode method. The modal aerodynamic force is obtained according to the first-order piston theory. The *p-k* method is introduced to calculate the flutter characteristics. Then, enhancement methods of the flutter velocity and flutter frequency of the panel with dampers are discussed. Therefore, as far as we know, there are two innovations in this paper that are not covered in other literature. First, based on the proposed assumed mode method, mass attachments are introduced into the structural model for aeroelastic analysis of thin-walled panels. The effects of the mass attachments on the inherent properties of the thin-walled panel are expressed analytically. Second, according to the developed aeroelastic model including mass attachments, flutter equations in the frequency domain of the panel are constructed. By analyzing the flutter characteristics (including flutter velocity and flutter frequency) of the panel, two reasonable preset damper configurations are proposed to recover and enhance the flutter characteristics of the panel with mass attachments. The measure can effectively improve the safety of aircraft structures. It should be noted that the paper uses the first-order piston aerodynamic theory and uses the *p-k* method to provide the flutter characteristics of the thin-walled panel in the frequency domain. The flutter calculation model used is linear. The influence of nonlinear factors (such as nonlinear aerodynamic force and nonlinear membrane force inside the panel) on the flutter characteristics of the panel is not considered.

## 2. Geometry of a Thin-Walled Panel with Mass Attachments

A three-dimensional thin-walled panel exposed to supersonic airflow that is simply supported on four sides is shown in Figure 1. The length of the panel along the airflow direction and the spanwise direction are $a$ and $b$, respectively. The thickness of the panel $h$ is uniform and much less than $a$ and $b$, so the Kirchhoff hypothesis [24] is satisfied. Since the supersonic airflow $U_\infty$ only flows on the outer surface, there is a pressure difference between the inner and outer surfaces of the panel. During a flight mission, mass attachments may exist on either the inner surface or the outer surface of the panel. In general, the areas of mass attachments are relatively smaller than that of the panel. Therefore, it is reasonable to consider mass attachments as several lumped masses in the dynamic model of the thin-walled panel. In addition, we assume that that there is no relative interface peeling and sliding between the panel and these mass attachments. In the established Cartesian coordinate system *Oxyz*, the coordinate origin *O* is located at a corner of the panel, the *x*-axis is the direction along the airflow, the *y*-axis is along the spanwise direction and the *z*-axis is the normal direction of the panel. The panel mainly deforms in the direction perpendicular to the surfaces when flutter occurs. Therefore, only the transverse vibration of the panel is considered.

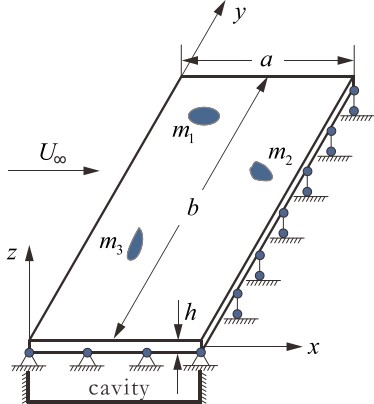

**Figure 1.** A simply supported panel with three mass attachments in the supersonic airflow.

### 3. Dynamic Equations Based on the Assumed Mode Method

According to the large displacement with small strain assumption proposed by von Kármán and ignoring the in-plane tensile deformation, the elastic potential energy of the panel shown in Figure 1 is

$$U_B = \frac{D_b}{2} \iint \left[ \left( \frac{\partial^2 w}{\partial x^2} \right)^2 + \left( \frac{\partial^2 w}{\partial y^2} \right)^2 + 2\mu \frac{\partial^2 w}{\partial x^2} \frac{\partial^2 w}{\partial y^2} + 2(1-\mu) \left( \frac{\partial^2 w}{\partial x \partial y} \right)^2 \right] \mathrm{d}x \mathrm{d}y \quad (1)$$

where $w$ is the transverse bending displacement of the panel, $\mu$ is Poisson's ratio, $D_b = Eh^3/12(1-\mu^2)$ is the bending stiffness of the panel and $E$ is the Young's modulus of the used material.

When there are no mass attachments on the thin-walled panel, the kinetic energy due to the bending deformation of the panel can be written as

$$T_B = \frac{1}{2} \iint \rho \left( \frac{\partial w}{\partial t} \right)^2 \mathrm{d}x \mathrm{d}y \quad (2)$$

where $\rho$ is the areal density of the panel. Therefore, the mass $m$ of the panel can be expressed as $m = \rho abh$. When the panel vibrates under supersonic airflow, the mass attachments shown in Figure 1 obtain additional kinetic energy, as

$$T_a = \frac{1}{2} \sum_{p=1}^{N_p} m_p \left( \frac{\partial w(x_p, y_p)}{\partial t} \right)^2 \quad (3)$$

where $N_p$ is the number of the mass attachments, $m_p$ is the mass of the $p$-th attachment and $x_p$ and $y_p$ are the $x$-coordinate and $y$-coordinate of the $p$-th mass attachment, respectively. Therefore, the total kinetic energy of the system can be written as the superposition of Equations (2) and (3):

$$T = \frac{1}{2} \iint \rho \left( \frac{\partial w}{\partial t} \right)^2 \mathrm{d}x \mathrm{d}y + \frac{1}{2} \sum_{p=1}^{N_p} m_p \left( \frac{\partial w(x_p, y_p)}{\partial t} \right)^2 \quad (4)$$

In the process of flutter calculation, modal properties of the panel have an important influence on its flutter characteristics. However, due to the existence of mass attachment, the modal properties of the simply supported panel on four sides have changed. In general, let the $i$-th order mode of the panel with mass attachments be $\phi_i(x,y)$ and the $i$-th order principal coordinate be $q_i(t)$, the transverse vibration displacement $w(x,y,t)$ of the panel can be expressed as

$$w(x,y,t) = \sum_{i=1}^{r} \phi_i(x,y) q_i(t) \quad (5)$$

where $r$ is the number of the truncated modes. However, unfortunately, with the addition of mass attachments, the modal properties of the panel become complex and change with the location $(x_p, y_p)$ and the number $N_p$ of attachments. In order to provide an analytical expression of $\phi_i(x,y)$, the modal function of the simply supported panel is taken as the assumed mode:

$$\widetilde{\phi}_i(x,y) = \sin \frac{m\pi x}{a} \sin \frac{n\pi y}{b}, m = 1, \cdots, N_m; \ n = 1, \cdots, N_n. \quad (6)$$

where $N_m$ and $N_n$ are the maximum number of modes of the panel in the $x$ and $y$ directions, respectively. The relationship between indexes $m$, $n$ and $i$ needs to be determined by the sequence of natural frequency values. By substituting Equation (6) into Equation (5), it

can be obtained that the transverse displacement of the panel expressed by the assumed mode is

$$\widetilde{w}(x,y,t) = \sum_{i=1}^{r} \widetilde{\phi}_i(x,y)q_i(t) \tag{7}$$

Substituting Equation (7) into Equation (1), the bending potential energy expression of the panel can be rewritten as

$$U_B = \frac{1}{2}\sum_{i=1}^{r}\sum_{j=1}^{r}\widetilde{k}_{ij}q_i(t)q_j(t) \tag{8}$$

where

$$\widetilde{k}_{ij} = D_b \iint \left[ \frac{\partial^2 \widetilde{\phi}_i}{\partial x^2}\frac{\partial^2 \widetilde{\phi}_j}{\partial x^2} + \frac{\partial^2 \widetilde{\phi}_i}{\partial y^2}\frac{\partial^2 \widetilde{\phi}_j}{\partial y^2} + 2\mu\frac{\partial^2 \widetilde{\phi}_i}{\partial x^2}\frac{\partial^2 \widetilde{\phi}_j}{\partial y^2} + 2(1-\mu)\frac{\partial^2 \widetilde{\phi}_i}{\partial x \partial y}\frac{\partial^2 \widetilde{\phi}_j}{\partial x \partial y} \right] dxdy \tag{9}$$

Similarly, by substituting Equation (7) into Equation (4), the total kinetic energy of the panel can be rewritten as

$$T = \frac{1}{2}\sum_{i=1}^{r}\sum_{j=1}^{r}\widetilde{m}_{ij}\dot{q}_i(t)\dot{q}_j(t) \tag{10}$$

where

$$\widetilde{m}_{ij} = \rho h \iint \widetilde{\phi}_i(x,y)\widetilde{\phi}_j(x,y)dxdy + \sum_{p=1}^{N_p} m_p \widetilde{\phi}_i(x_p,y_p)\widetilde{\phi}_j(x_p,y_p) \tag{11}$$

Let $\widetilde{K} = \left[\widetilde{k}_{ij}\right]_{r\times r}$ and $\widetilde{M} = \left[\widetilde{m}_{ij}\right]_{r\times r}$ and solve the generalized eigenvalue problem

$$\left|\widetilde{K} - \omega^2 \widetilde{M}\right| = 0 \tag{12}$$

Then, the first $r$-th order natural frequencies and eigenvectors of the system with mass attachments can be obtained as $\omega_i, i = 1, \cdots, r$ and $a_1, \cdots, a_r$, respectively. Furthermore, let $\mathbf{\Phi} = [a_1, \cdots, a_r]_{r\times r}$, $\boldsymbol{\phi} = [\phi_1, \cdots, \phi_r]$ and $\widetilde{\boldsymbol{\phi}} = [\widetilde{\phi}_1, \cdots, \widetilde{\phi}_r]$, the relationship between the assumed modes and the approximate modes of the panel with mass attachments is

$$\boldsymbol{\phi} = \widetilde{\boldsymbol{\phi}}\mathbf{\Phi} \tag{13}$$

Then, the $i$-th order mode function of the panel with mass attachments is

$$\phi_i(x,y) = \sum_{j=1}^{r} \widetilde{\phi}_j(x,y)a_{ji} \tag{14}$$

Substituting Equation (14) into Equation (5), the transverse vibration displacement $w(x,y,t)$ of the panel can be rewritten as

$$w(x,y,t) = \sum_{i=1}^{r}\sum_{j=1}^{r} \widetilde{\phi}_j(x,y)a_{ji}q_i(t) \tag{15}$$

By introducing parameters

$$M_i = \iint \rho h[\phi_i(x,y)]^2 dxdy = \iint \rho h \left[\sum_{j=1}^{r}\widetilde{\phi}_j(x,y)a_{ji}\right]^2 dxdy \tag{16}$$

and

$$K_i = D_b \iint \left[ \frac{\partial^2 \phi_i}{\partial x^2} \frac{\partial^2 \phi_i}{\partial x^2} + \frac{\partial^2 \phi_i}{\partial y^2} \frac{\partial^2 \phi_i}{\partial y^2} + 2\mu \frac{\partial^2 \phi_i}{\partial x^2} \frac{\partial^2 \phi_i}{\partial y^2} + 2(1-\mu) \frac{\partial^2 \phi_i}{\partial x \partial y} \frac{\partial^2 \phi_i}{\partial x \partial y} \right] dx dy \qquad (17)$$

here since the stiffness of the panel does not change, the assumed modes can also be used to calculate $K_i$ as:

$$K_i = D_b \iint \left[ \frac{\partial^2 \widetilde{\phi}_i}{\partial x^2} \frac{\partial^2 \widetilde{\phi}_i}{\partial x^2} + \frac{\partial^2 \widetilde{\phi}_i}{\partial y^2} \frac{\partial^2 \widetilde{\phi}_i}{\partial y^2} + 2\mu \frac{\partial^2 \widetilde{\phi}_i}{\partial x^2} \frac{\partial^2 \widetilde{\phi}_i}{\partial y^2} + 2(1-\mu) \frac{\partial^2 \widetilde{\phi}_i}{\partial x \partial y} \frac{\partial^2 \widetilde{\phi}_i}{\partial x \partial y} \right] dx dy \qquad (18)$$

The first $r$ order natural frequencies of the panel are

$$\omega_i = \sqrt{\frac{K_i}{M_i}}, i = 1, \cdots, r. \qquad (19)$$

In addition, the mass matrix and stiffness matrix of the panel can be written as $\boldsymbol{M} = [M_i]_{r \times r}$ and $\boldsymbol{K} = [K_i]_{r \times r}$, respectively. Thus, the system motion equation represented by the modal coordinate can be obtained as

$$M_i \ddot{q}_i(t) + C_i \dot{q}_i(t) + K_i q_i(t) = Q_i(t), i = 1, 2, \cdots, r \qquad (20)$$

where $Q_i$ is the $i$-th order mode aerodynamic force, which will be determined in Section 4, and $C_i$ is the modal damping coefficient which is usually composed of two parts, $C_i^{\text{I}}$ and $C_i^{\text{O}}$. $C_i^{\text{I}}$ is the structural damping caused by the internal friction in the vibration process and $C_i^{\text{O}}$ is the effect caused by several additional artificial dampers. The relationship between them is

$$C_i = C_i^{\text{I}} + C_i^{\text{O}} \qquad (21)$$

Let $\boldsymbol{C} = [C_i]_{r \times r}$ and $\boldsymbol{Q} = [Q_i]_{r \times r}$, the structural dynamic equation in the modal coordinate system of the panel can be written in matrix form as

$$\boldsymbol{M}\ddot{\boldsymbol{q}}(t) + \boldsymbol{C}\dot{\boldsymbol{q}}(t) + \boldsymbol{K}\boldsymbol{q}(t) = \boldsymbol{Q}(t) \qquad (22)$$

## 4. Aerodynamic Force in Modal Coordinate System

In the case of the zero air deflection angle, the supersonic flow on the outer surface of the panel flows along the positive direction of the $x$-axis. At this time, the $z$-direction velocity of a point on the outer surface of the panel is

$$v_z = \left( U_\infty \frac{\partial}{\partial x} + \frac{\partial}{\partial t} \right) w(x, y, t) = \left( U_\infty \frac{\partial}{\partial x} + \frac{\partial}{\partial t} \right) \left( \sum_{i=1}^{r} \sum_{j=1}^{r} \widetilde{\phi}_j(x, y) a_{ji} q_i(t) \right) \qquad (23)$$

Introducing the 1st order piston theory, the pressure difference $\Delta p(x, y, t)$ between the outer and inner surface of the panel can be expressed as

$$\Delta p(x, y, t) = -\frac{2q_d}{Ma_\infty} \left( \frac{\partial w(x, y, t)}{\partial x} + \frac{1}{U_\infty} \frac{\partial w(x, y, t)}{\partial t} \right) \qquad (24)$$

where $q_d$ is the dynamic pressure and $Ma_\infty$ is the Mach number of the undisturbed air. By substituting Equation (15) into Equation (24), we can rewrite the pressure difference $\Delta p(x, y, t)$ by exchanging the order of the sum and derivative operations as

$$\Delta p(x, y, t) = -\frac{2q_d}{Ma_\infty} \left( \sum_{i=1}^{r} \sum_{j=1}^{r} \frac{\partial \widetilde{\phi}_j(x, y)}{\partial x} a_{ji} q_i(t) + \frac{1}{U_\infty} \sum_{i=1}^{r} \sum_{j=1}^{r} \widetilde{\phi}_j(x, y) a_{ji} \dot{q}_i(t) \right) \qquad (25)$$

Then, the *i*-th modal aerodynamic force $Q_i$ can be written as

$$Q_i = \iint \Delta p(x,y,t)\phi_i(x,y)\mathrm{d}x\mathrm{d}y = -\frac{2q_d}{Ma_\infty}\sum_{j=1}^{r}\left(\overline{D}_{ij}q_j(t) + \frac{1}{U_\infty}\overline{E}_{ij}\dot{q}_j(t)\right) \tag{26}$$

where

$$\overline{D}_{ij} = \iint\left[\left(\sum_{s=1}^{r}\widetilde{\phi}_s(x,y)a_{si}\right)\left(\sum_{s=1}^{r}\frac{\partial\widetilde{\phi}_s(x,y)}{\partial x}a_{sj}\right)\right]\mathrm{d}x\mathrm{d}y \tag{27}$$

$$\overline{E}_{ij} = \iint\left[\left(\sum_{s=1}^{r}\widetilde{\phi}_s(x,y)a_{si}\right)\left(\sum_{s=1}^{r}\widetilde{\phi}_s(x,y)a_{sj}\right)\right]\mathrm{d}x\mathrm{d}y \tag{28}$$

It can be seen from Equation (26) that the aerodynamic coefficients acting on the panel consist of two parts. One part is the coefficient $\overline{D}_{ij}$ in front of $q_j(t)$, which plays the role of aerodynamic stiffness. The other part is the coefficient $\overline{E}_{ij}$ in front of $\dot{q}_j(t)$, which plays the role of aerodynamic damping.

It should be noted that in Equation (24), $Ma_\infty$ is generally required to be no less than 2.0. Single mode flutter occurring at $Ma_\infty < 2$ cannot be detected by the use of piston theory. This type of flutter is not considered in this work. The papers of Abdukhakimov et al. [25] and Shishaeva et al. [26] can be discussed in this regard.

## 5. Flutter Equation

By substituting Equation (26) into Equation (20), the aeroelastic system motion equation of the panel with mass attachments can be represented in the modal coordinate as

$$M_i\ddot{q}_i + C_i\dot{q}_i + K_iq_i = -\frac{2q_d}{Ma_\infty}\sum_{j=1}^{n}\left(\overline{D}_{ij}q_j(t) + \frac{1}{U_\infty}\overline{E}_{ij}\dot{q}_j(t)\right), i = 1, 2, \cdots, r \tag{29}$$

Equation (29) can be rewritten in matrix form as

$$\boldsymbol{M}\ddot{\boldsymbol{q}}(t) + \boldsymbol{C}\dot{\boldsymbol{q}}(t) + \boldsymbol{K}\boldsymbol{q}(t) = -\frac{2q_d}{Ma_\infty}\left(\overline{\boldsymbol{D}}\boldsymbol{q}(t) + \frac{1}{U_\infty}\overline{\boldsymbol{E}}\dot{\boldsymbol{q}}(t)\right) \tag{30}$$

Equation (30) is the time domain form of the flutter equation of the panel with mass attachments based on the 1st order piston theory. In order to understand the flutter trend of the panel in the subcritical state, the *p-k* method is introduced to calculate the flutter characteristics of the panel with mass attachments and artificial dampers. Therefore, the modal displacement vector $\boldsymbol{q}(t)$ is rewritten as follows

$$\boldsymbol{q}(t) = \overline{\boldsymbol{q}}e^{st} \tag{31}$$

where $\overline{\boldsymbol{q}}$ is the amplitude of $\boldsymbol{q}(t)$ and is independent of time $t$, and

$$s = (\gamma + \mathrm{i})\omega \tag{32}$$

where $\gamma$ is the transient decay rate coefficient, which determines the speed of vibration attenuation. By introducing a dimensionless parameter

$$p = (\gamma + 1)k \tag{33}$$

where $k$ is the reduced frequency and $\omega = 2kU_\infty/a$ for the panel with mass attachments and artificial dampers, Equation (31) can be rewritten as

$$\boldsymbol{q}(t) = \overline{\boldsymbol{q}}e^{2pU_\infty t/a} \tag{34}$$

By substituting Equation (34) into Equation (29) and letting

$$\overline{A} = \left(\frac{2U_\infty}{a}\right)^2 M \tag{35}$$

$$\overline{B} = \frac{2U_\infty}{a}C - q_d\frac{\mathbf{Q}_{\mathrm{Im}}(k, Ma_\infty)}{k} \tag{36}$$

$$\overline{C} = K - q_d\mathbf{Q}_{\mathrm{Re}}(k, Ma_\infty) \tag{37}$$

where $\mathbf{Q}(k, Ma_\infty) = -\frac{2}{Ma_\infty}\left(\overline{D} + \frac{2p}{a}\overline{E}\right)$ and $Ma_\infty$ is the Mach number during flight, we can obtain the quadratic eigenvalue problem of the panel aeroelastic system with mass attachments and artificial dampers as follows:

$$\left[\overline{A}p^2 + \overline{B}p + \overline{C}\right]\overline{q} = 0 \tag{38}$$

The calculation of $p$ in Equation (38) requires an iterative process. When the system converges at the current flow rate $U_\infty$, we can obtain the structural damping $g$ and frequency $f$(Hz) as

$$g \approx 2\gamma = 2\frac{\mathrm{Re}(p)}{k} \tag{39}$$

$$f = \frac{kU_\infty}{a\pi} \tag{40}$$

## 6. Numerical Example

The square panel in a wing structure shown in Figure 2 is simply supported on four sides when flying at supersonic speed. The side length is $a = b = 300$ mm and the thickness is $h = 1.2$ mm. The physical parameters of the panel are shown in Table 1 [27,28]. In the preliminary flutter calculation process, it is considered that the panel has no mass attachment, as shown in Figure 2a. Then, we start from Section 6.1 to consider the case of several mass attachments attached to the panel, as shown in Figure 2b. Generally, the shapes of the mass attachments are irregular. However, since their characteristic sizes are much smaller than the surface area of the wall panel, it is generally considered as concentrated mass points in this paper. The positive direction of the $x$-axis is the direction of the airflow. When the aircraft is flying near sea level, the air density is taken as $\rho_\infty = 1.226$ kg/m$^3$, and the air density ratio is taken as $\rho_r = 1.0$.

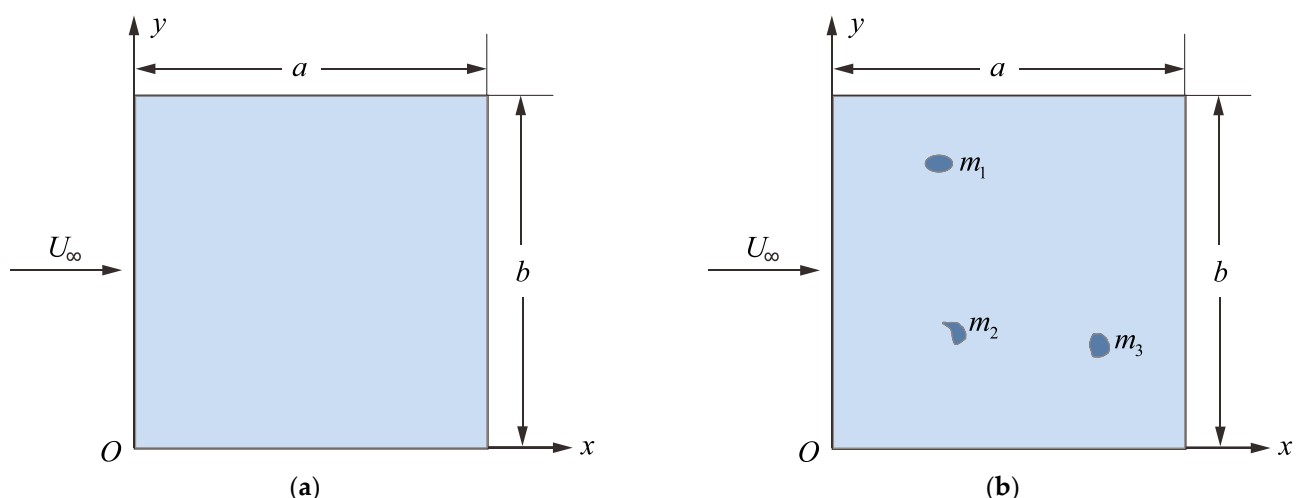

**Figure 2.** A square panel with simply supported boundaries on four sides: (**a**) with no mass attachment; (**b**) with several mass attachments.

**Table 1.** Physical parameters of the square panel.

| Physical Quantity | Value |
|---|---|
| Young's modulus (Pa) | $E = 7.1 \times 10^{10}$ |
| Poisson's ratio | $v = 0.32$ |
| Mass density (kg/m3) | $\rho_s = 2768$ |
| Modal damping ratio (%) | $\zeta = 0.0$ |

Figure 3 shows the variation in the flutter velocities and flutter frequencies of the panel with the number of truncated modes when $Ma_\infty = 2.0$. It can be seen from Figure 3 that at least the first 15 natural modes of the panel structure should be taken to participate in the flutter calculation in order to obtain the correct flutter velocity and frequency. Therefore, the first 16 modes are taken to participate in the flutter calculation of the panel in order to reduce the calculation efforts while ensuring the calculation accuracy. So, taking the first 16 order modes (namely $N_m = 4$, $N_n = 4$ in Equation (6)) of the panel to participate in the flutter calculation and ignoring the structural damping, the nonmatching flutter velocity of the panel is calculated under $Ma_\infty = 2.0$, as shown in Figure 4. It can be seen from Figure 4 that the flutter velocity and flutter velocity calculated by the program proposed in this paper are $U_F = 543.4$ m/s and $f_F = 133.9$ Hz, respectively. Flutter occurs in the second mode. It can be seen from Figure 4b that the first mode and the second mode are coupled when flutter occurs. In addition, we can also obtain the slope of the third mode when $g = 0$ in Figure 4a as an index to evaluate the suddenness of entering flutter. Here, the index is called the flutter slope and is denoted as $s_F$. Here, $s_F = 0.0155$ in the case of no mass attachment. As a comparison, the same problem can be solved with the Nastran code. The results using the Nastran software (*p-k* method) are $U_F = 541.2$ m/s, $f_F = 133.5$ Hz [27]. The flutter velocity and the flutter frequency differ by 0.41% and 0.30%, respectively, and the flutter shape is the same as that calculated in this paper. For convenience, in the following sections, we denote the case without mass attachment as NP.

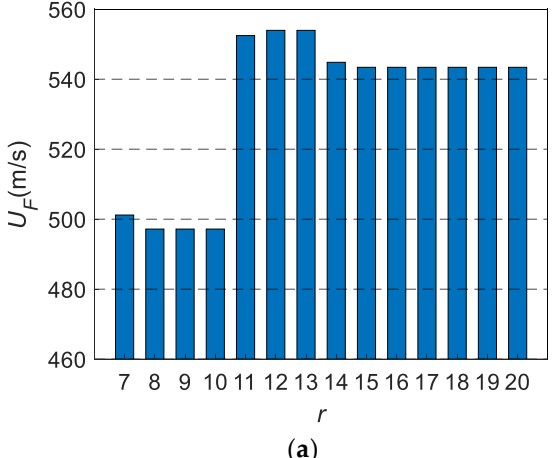

(a)

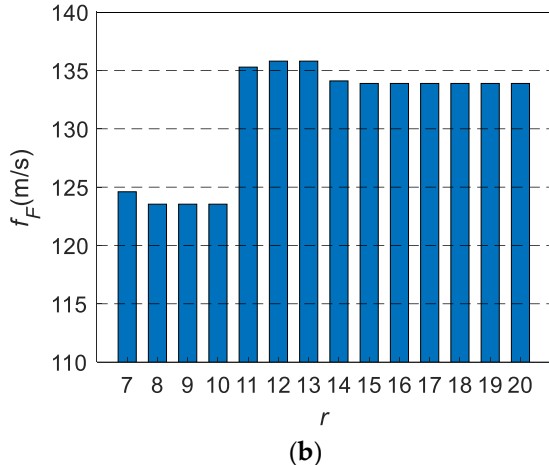

(b)

**Figure 3.** Flutter velocities and flutter frequencies change with the number of truncated modes $r$ ($Ma_\infty = 2.0$): (**a**) flutter velocities, (**b**) flutter frequencies.

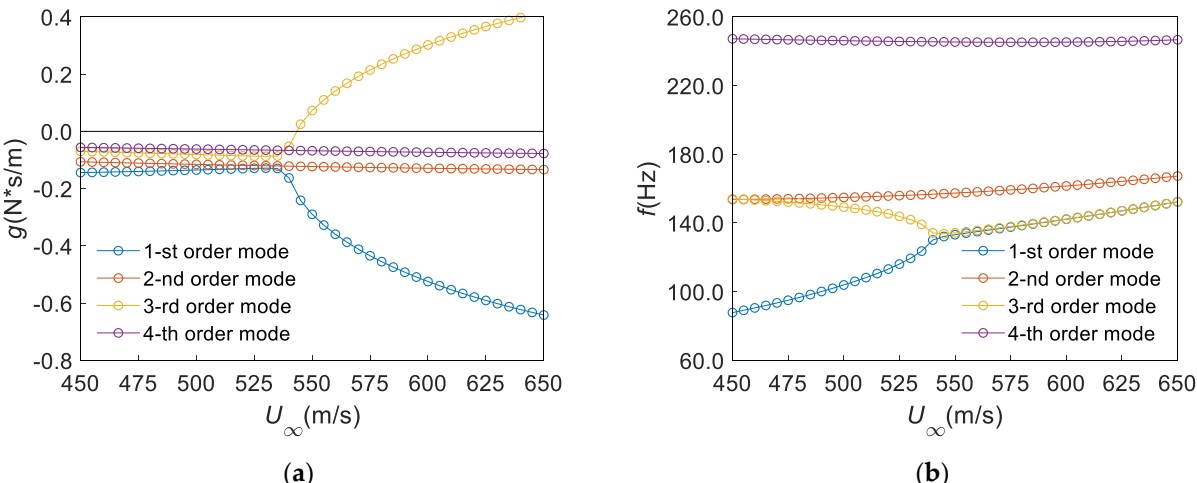

**(a)**                    **(b)**

**Figure 4.** Results of flutter calculation for the panel with no mass attachment (NP) ($Ma_\infty = 2.0$):
(**a**) $U - g$, (**b**) $U - f$.

### 6.1. The Case of Only One Mass Attachment

In this section, we consider the case of only one mass attachment on the panel. The mass of the attachment is 0.03 kg (about 1/10 of the mass of the panel). When it is located at different positions, it will have different effects on the flutter characteristics of the panel. To this end, we first consider two situations. One is that the mass attachment is located at the center of the panel, as shown in Figure 5a. The other is that the mass attachment is located at the intersection point of 1/4 chord length and 1/4 span, as shown in Figure 5b. For convenience, the two different cases are denoted as CP and QP, respectively. Using the method proposed in this paper, the flutter velocities and flutter frequencies of the panel under the two cases are calculated, as shown in Figures 6 and 7.

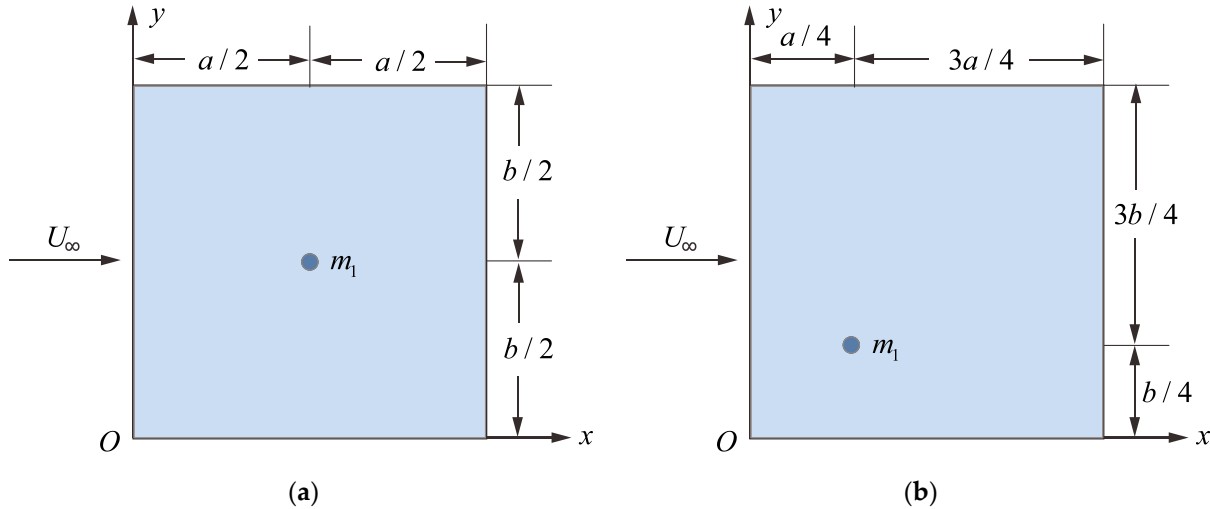

**(a)**                    **(b)**

**Figure 5.** Two different positions of a mass attachment: (**a**) the center point of the panel (CP); (**b**) the intersection point located at 1/4 span length and 1/4 chord length (QP).

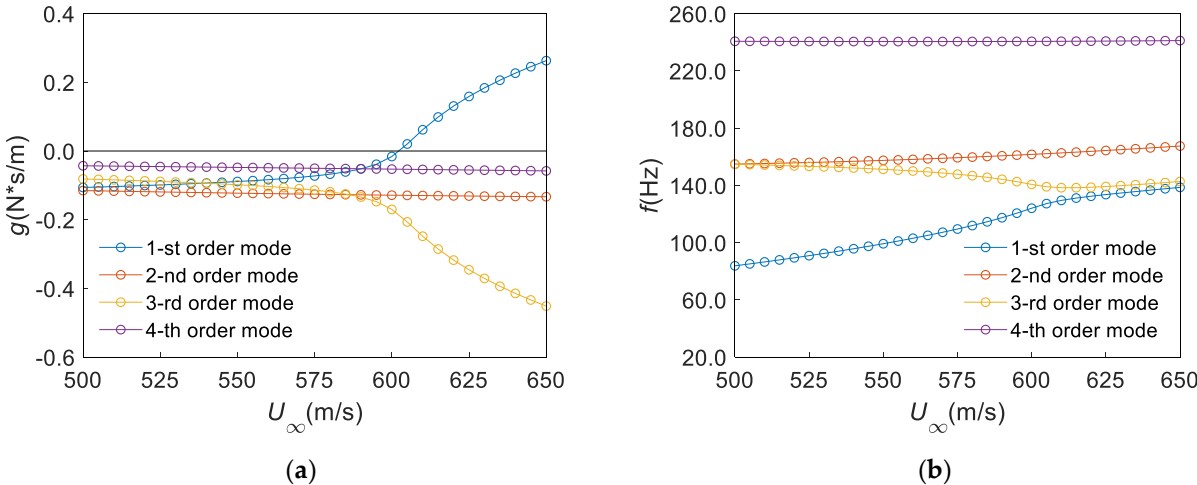

**Figure 6.** Results of flutter calculation (CP): (**a**) $U - g$, (**b**) $U - f$.

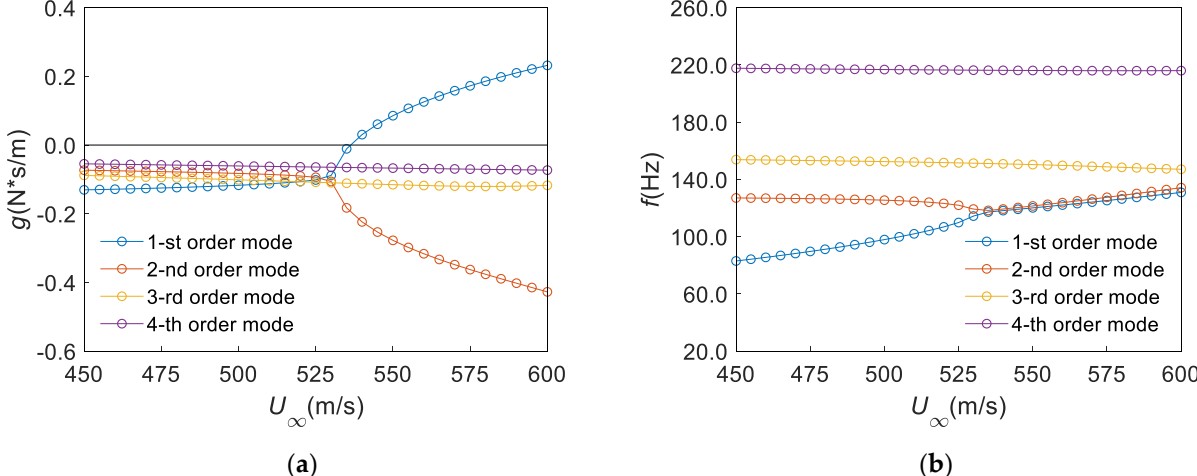

**Figure 7.** Results of flutter calculation (QP): (**a**) $U - g$, (**b**) $U - f$.

It can be seen from Figure 5 that the flutter velocity and the flutter frequency in the case of CP are 602.2 m/s and 125.4 Hz, respectively. It should be noted that the flutter mode jumps from the second mode to the first mode, but when flutter occurs, the first mode and the second mode are also coupled. Compared with the case of NP, the flutter velocity was increased by about 10.82%. However, the flutter frequency decreased by 6.35%. Additionally, the flutter slope became 0.0073 and the suddenness decreased by 52.90%.

When the mass attachment is located at QP, the corresponding flutter velocity and the flutter velocity as shown in Figure 7 are 536.3 m/s and 117.5 Hz, respectively. Flutter occurs again at the first mode. When flutter occurs, the first mode is coupled with the second mode. At the same time, the flutter slope can be obtained as 0.0083, which is slightly higher than the case of CP, but still 46.5% smaller than the case of NP.

From Figures 6 and 7, it can be concluded that when a mass attachment is added to the panel, the change in the flutter characteristics (flutter velocity, flutter frequency and flutter slope) is closely related to the position of the mass attachment. In particular, the flutter velocity may increase or decrease. This is actually a challenge to the design of the panel structure, which needs to be taken seriously.

In order to further understand the influence of the location change in the mass attachment on the natural properties and flutter characteristics of the panel, we drew the changes in the first four natural frequencies with the location $(x, y)$ of the mass attachment in Figure 8. It can be seen from Figure 8 that the first, second and fourth natural frequencies

$f_1$, $f_2$ and $f_4$ of the panel all decreased to varying degrees due to the mass gain brought by the attachment. However, the laws of decreasing are different. It can be seen from Figure 8a that the closer the mass attachment is to the center of the panel, the lower the first order natural frequency. Compared with the case of NP, the maximum decreasing amplitude is up to 10.29 Hz. In addition, we can also observe that there is only one minimum point (namely the center point) in the first order natural frequency image. However, both the second order and the fourth order natural frequency images evolve into four extreme regions. The difference is that there is a connected area in the $f_2$ image, while the $f_4$ image is divided into four independent areas by the symmetry axis of the panel. It can be observed from Figure 8b that when the mass attachment is located near the center of the panel, the second order natural frequency has a higher value. Additionally, it can be seen from Figure 8d that when the mass attachment is near the symmetry axis of the panel, the fourth order natural frequency is almost equal to that in the case of NP.

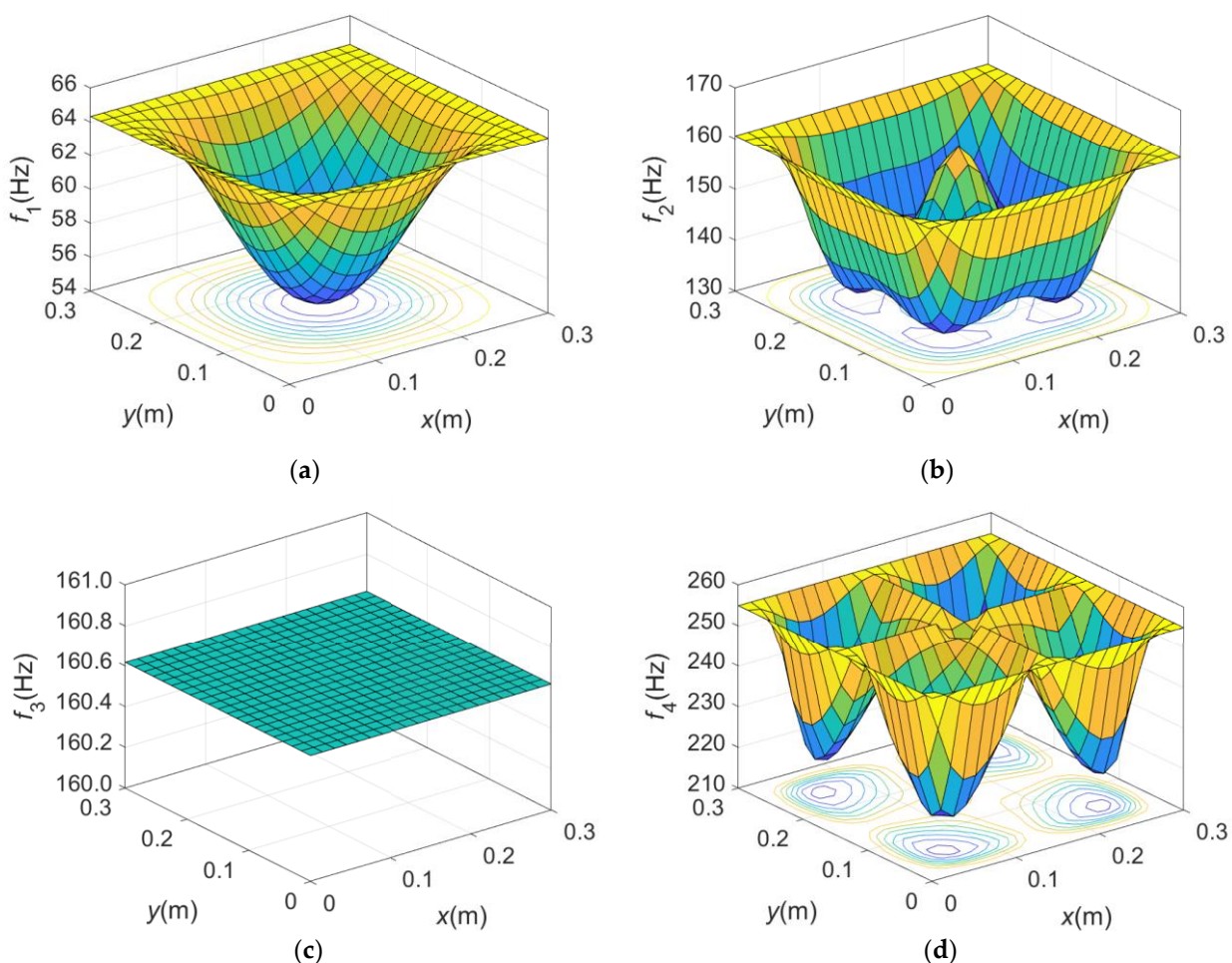

**Figure 8.** Graphs of the change in the first four natural frequencies of the panel with the position $(x, y)$ of the mass attachment: (**a**) the first order natural frequency $f_1$; (**b**) the second order natural frequency $f_2$; (**c**) the third order natural frequency $f_3$; (**d**) the fourth order natural frequency $f_4$.

Interestingly, it can be seen from Figure 8c that the value of the third order natural frequency $f_3$ remains constant no matter how the position $(x, y)$ of the mass attachment changes. Further study shows that this phenomenon also occurs at the 8th and 10th natural frequencies. The reason is that when solving the generalized eigenvalue problem, the integral value of the modes corresponding to these frequencies in the domain is 0, which is related to the orthogonality of the modes.

Furthermore, we drew the changes in the flutter characteristics of the panel with the location $(x, y)$ of the mass attachment in Figure 9. It can be seen from Figure 8a that the variation in the flutter velocity $U_F$ with the position $(x, y)$ of the mass attachment is complex. In the central region of the panel, the flutter velocity shows a large gradient trend. At the center point of the panel, $U_F$ is at the maximum state. However, at a slight deviation from the center point, $U_F$ drops sharply to the minimum point. The variation amplitude of the flutter velocities reaches 87.7 m/s. However, a different variation pattern appears in the flutter frequency image, as shown in Figure 9b. The flutter frequencies of the panel decrease compared with the NP case, except that the mass attachment is located on the central axis of the panel in the direction of air flow and is not near the central area of the panel. Similar to the case of the flutter velocity, the flutter frequency also changes dramatically, with a difference of 37.5 Hz between the maximum and minimum values.

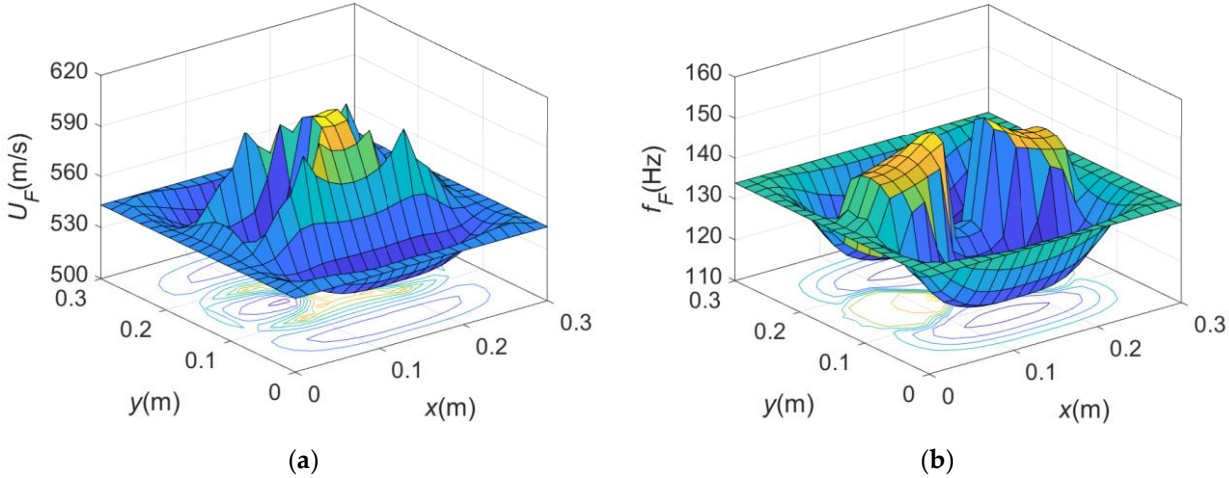

(a)   (b)

**Figure 9.** The changes in the flutter characteristics of the panel with the location $(x, y)$ of the mass attachment: (**a**) flutter velocity $U_F$, (**b**) flutter frequency $f_F$.

On the other hand, in order to investigate the changes in the flutter characteristics caused by the change in the attachment mass, the changes in flutter speed $U_F$ and flutter frequency $f_F$ with $m_1$ under the case of CP and QP are shown in Figure 10. It can be seen from Figure 10 that under the case of CP, both the flutter velocities and flutter frequencies have transitions at $m_1 = 0.1$. This is due to the flutter mode jump. When $m_1 < 0.1$, the flutter velocity increases linearly with the increase in $m_1$. After the jump, $U_F$ decreases monotonously and shows a nonlinear trend. Until $m_1 = 0.3$ (equivalent to the mass of the panel $m$), the flutter velocity decreases to 459.3 m/s. Compared with the case of NP, the flutter velocity decreases by 84.1 m/s, and the change rate reaches 15.48%. However, it can be observed from Figure 10b that the changes in flutter frequencies are different to those of flutter velocities. Flutter frequencies decrease with the increase in $m_1$ whether before or after the flutter mode jumps. The gain in the flutter frequency due to the mode jump reaches 87.0 Hz (from 108.3 Hz to 195.3 Hz).

Different from the case of CP, the flutter mode jump occurs near $m = 0.05$ in the case of NP. The flutter velocity is nonmonotonic when $m < 0.05$, and the minimum value appears near $m = 0.0025$. In contrast, flutter frequencies decrease monotonically and basically change linearly. However, flutter velocities and frequencies become insensitive to the change in $m_1$ after the flutter mode jumps.

That is, when the mass of the attachment on the panel changes (such as the accumulation of the ice load on the panel), the flutter characteristics of the panel will change greatly due to the jump in the flutter mode. These will also bring a serious threat to the safety of the panel structure and bring challenges to the design of the panel structure.

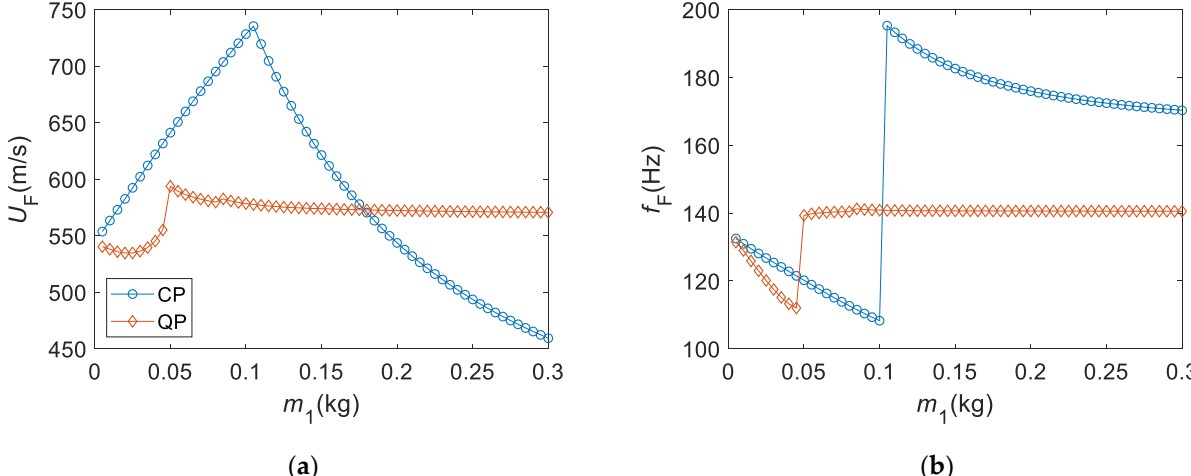

(**a**)

(**b**)

**Figure 10.** The changes in flutter characteristics caused by the change in attachment mass: (**a**) flutter velocity $U_F$, (**b**) flutter frequency $f_F$.

### 6.2. The Case of Two or More Mass Attachments

When there are two or more mass attachments on the panel, each attachment contributes to the quality attributes of the panel. The inherent properties of the panel will change under the joint influence of these attachments and then affect the panel's flutter characteristics. Only two new cases are considered here to avoid complexity, as shown in Figure 11. The first case is the combination of the case of CP and the case of QP, here called the case TP, as shown in Figure 11a. The other case, which is called MP, includes five mass attachments, whose positions are shown in Figure 11b.

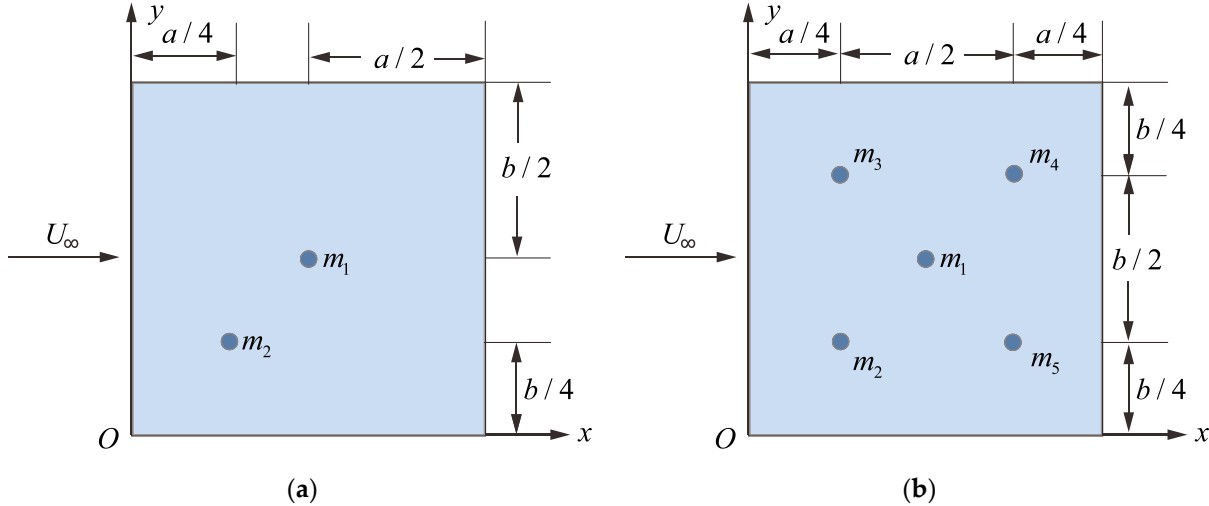

(**a**)

(**b**)

**Figure 11.** Two position relations of the panel with two or more mass attachments: (**a**) one point located at CP and the other at QP (TP); (**b**) there are 5 mass attachments on the panel (MP).

The flutter velocities and flutter frequencies of the panel in the case of TP and MP are shown in Figures 12 and 13, respectively. It can be seen from Figure 12 that the flutter velocity of the panel is 574.0 m/s, the flutter frequency is 111.8 Hz and the flutter slope is 0.0119. The flutter occurs in the first mode of the panel. When flutter occurs, the first mode is coupled with the second mode. Similarly, it can be seen from Figure 13 that the flutter velocity of the panel is 537.1 m/s, the flutter frequency is 97.92 Hz and the flutter slope is 0.0155. However, the flutter occurs in the second mode of the panel, which is coupled with the first mode when flutter occurs.

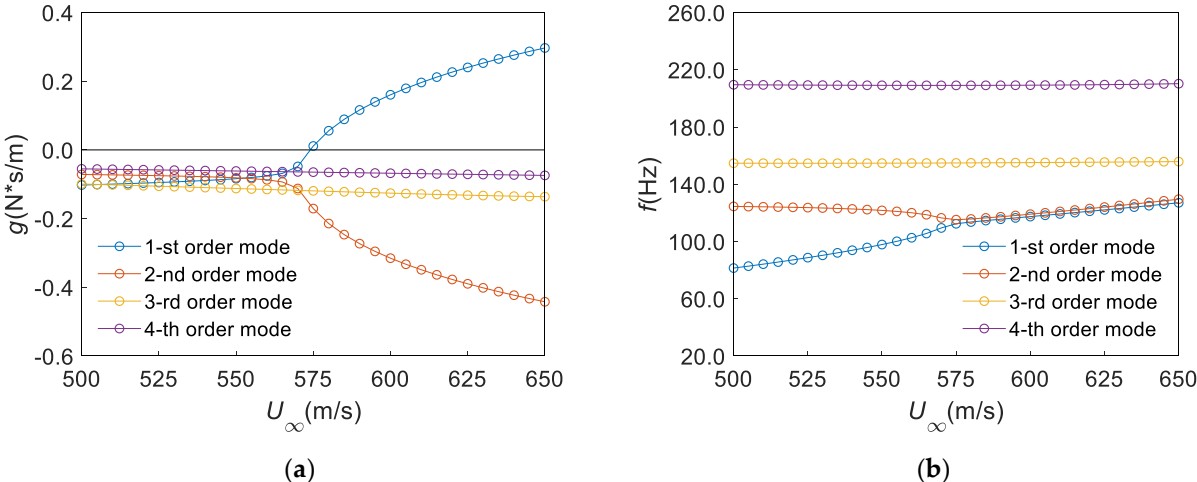

**Figure 12.** Results of flutter calculation for the panel with two mass attachments (TP) ($Ma_\infty = 2.0$): (**a**) $U - g$, (**b**) $U - f$.

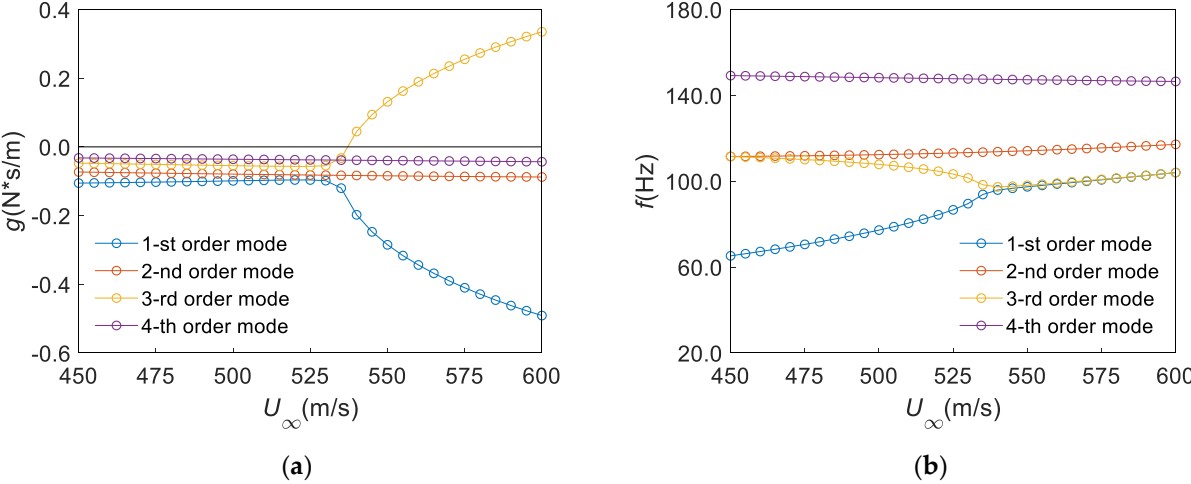

**Figure 13.** Results of flutter calculation for the panel with five mass attachments (MP) ($Ma_\infty = 2.0$): (**a**) $U - g$, (**b**) $U - f$.

A comparison of the flutter characteristics of the panel under different cases (NP, CP, QP, TP and MP) are listed in Table 2. It can be seen from Table 2 that the flutter velocity in the case of TP is between that in the case of CP and QP. This is because the existence of a mass attachment can increase the flutter velocity of the panel in the case of CP, while it decreases the flutter velocity in the case of QP. Their contributions to the flutter velocity are reconciled when the two mass attachments exist. As the number of mass attachments increases, the flutter velocity of the panel continues to decrease. Coincidentally, the flutter velocity under the case of MP is roughly the same as that under the case of QP.

**Table 2.** Comparison of flutter characteristics of the panel under different cases.

| Case | $U_F$ | $f_F$ | $s_F$ |
|------|-------|-------|-------|
| NP | 543.4 | 133.9 | 0.0155 |
| CP | 602.2 | 125.4 | 0.0073 |
| QP | 536.3 | 117.5 | 0.0083 |
| TP | 574.0 | 111.8 | 0.0012 |
| MP | 537.1 | 97.92 | 0.0155 |

Different from the flutter velocity, the flutter frequency in the case of MP is the minimum in all the considered cases. This means that as the number of mass attachments on the panel increases, the flutter frequency of the panel gradually decreases. Unfortunately, we cannot draw conclusion that the more mass attachments there are and the greater the mass, the lower the suddenness of entering the flutter state. In fact, the suddenness of entering the flutter state is related to the mass distribution of the panel.

### 6.3. Flutter Characteristics with Dampers

According to the results in Sections 6.1 and 6.2, we know that the flutter characteristics, especially the flutter velocities of the panel, may be decreasing or increasing depending on the positions and masses of attachments. However, the flutter velocity envelope is determined when the design of the panel is finalized. In the subsequent use, the inherent properties of the panel will change if there are mass attachments on the panel. Additionally, the flutter velocity and frequency will also change. It will seriously threaten the safety of personnel and aircraft if the flutter velocity of the panel drops too much. Therefore, some measures should be taken to maintain or even improve the flutter characteristics even when there are mass attachments on the panel after the panel is designed and finalized. It can be seen from Equation (38) that the existence of damping in the structural system can effectively change the solutions of the eigenvalue problem. This is because the damping in the structure will constantly absorb the energy generated by the self-excited vibration of the panel, so that a larger flow speed is required to maintain the simple harmonic motion state. So, it is a feasible solution to place some dampers at the position where the vibration velocity is the largest. According to the deformation theory of the Kirchhoff plate, the lower-order main vibrations $q_i(t), (i = 1, 2, 3, \cdots)$ of the panel are dominant when flutter occurs. The corresponding vibration modes are also the main components of the panel deformation. Therefore, for the $i$-th order main vibration $q_i(t)$ of the panel, the velocity maximum point actually corresponds to the displacement maximum point on the mode shape. That is to say, we can obtain the position of the damper required by analyzing the position of the maximum relative displacement point on the lower-order vibration modes. However, the trouble is that it is difficult for us to determine the locations, numbers and masses of attachments in advance during the use of a panel. Fortunately, it can be seen from Equation (14) that the vibration mode of the panel with mass attachments can be obtained by the superposition of the vibration modes of the panel under the case of NP. Therefore, as a preset method, we can configure corresponding damping points under the case of NP in advance and expect them to effectively improve the flutter characteristics of the panel in the presence of mass attachments. Based on this idea, we provided the first four mode shapes of the panel structure under the case of NP in Figure 14.

As can be seen from Figure 14, there are five displacement extreme points in the first three modes and nine in the first four modes. On the other hand, according to the flutter analysis in Sections 6.1 and 6.2, the flutter of panels with and without mass attachments occurs in the first three modes. Therefore, we propose two damper configurations to suppress panel flutter. One is arranging the dampers at the positions determined by the maximum displacement points from the first mode to the third mode, while the other is arranging the dampers at the positions determined by the maximum displacement points from the first mode to the fourth mode. The two damper configurations are called the 5P method and 9P method, respectively as shown in Figure 15.

In practical engineering applications, the boundary of the panel is often firmly connected to the frame, beam, rib or stringer through rivets, bolts, etc. Points on the boundary of the panel are generally considered to be in a simply supported state with displacements of 0. When an internal point of the panel is displaced under the action of air flow, an additional displacement difference is generated between a boundary point and the internal point of the panel. A velocity difference is then generated, which can be used to design a damper. Therefore, a light linear damper can be arranged between a damping configuration point of the panel and a boundary point or a panel corner point in advance, as shown in

Figure 16. In Figure 16a, the artificial damping coefficients $c_2$, $c_3$, $c_4$ and $c_5$ are all realized by setting a linear damper between the damping configuration point and a boundary point. Additionally, the damping coefficient $c_1$ can be achieved by connecting the corner points of the panel with four dampers. However, in the configuration of the 9P method, the additional artificial damping coefficients $c_6$, $c_7$, $c_8$ and $c_9$ can be achieved by connecting the damping configuration points and the four corner points, as shown in Figure 16b.

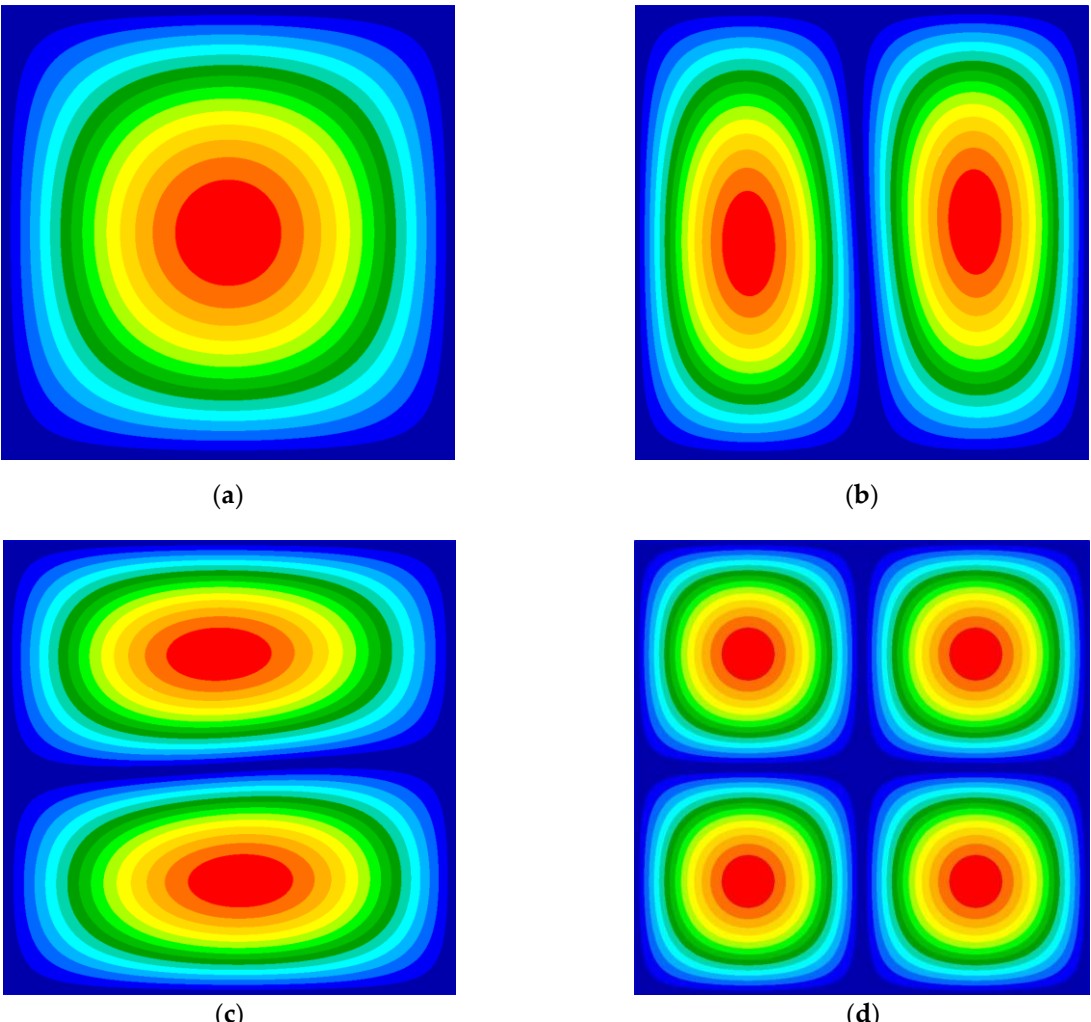

(a)

(b)

(c)

(d)

**Figure 14.** The first 4 modes of the panel with no mass attachment: (**a**) the 1st mode (64.55 Hz); (**b**) the 2nd mode (161.4 Hz); (**c**) the 3rd mode (161.4 Hz); (**d**) the 4th mode (258.1 Hz).

The case of QP is taken as an example to show the enhancement effects of the flutter characteristics of the panel with dampers. In this paper, only a simple case is considered, that is, the damping coefficients of the dampers arranged in advance are the same. First, let the damping coefficient $c = 1$ to obtain the flutter velocities and flutter frequencies of the panel in the case of QP calculated based on the 5P method and the 9P method, respectively, as shown in Figures 17 and 18. It can be seen from Figure 17 that the flutter velocity of the panel based on the 5P method is 541.9 m/s, and the flutter frequency is 118.7 Hz. The flutter occurs in the second mode of the panel. When flutter occurs, the second mode is coupled with the first mode. However, it can be seen from Figure 17 that the flutter velocity of the panel is 545.2 m/s and the flutter frequency is 118.5 Hz. The flutter occurs in the first mode of the panel which is coupled with the second mode when flutter occurs. By comparing the flutter frequency curves shown in Figures 17b and 18b, it can be seen that the results obtained by the five-point method and the nine-point method are basically the same.

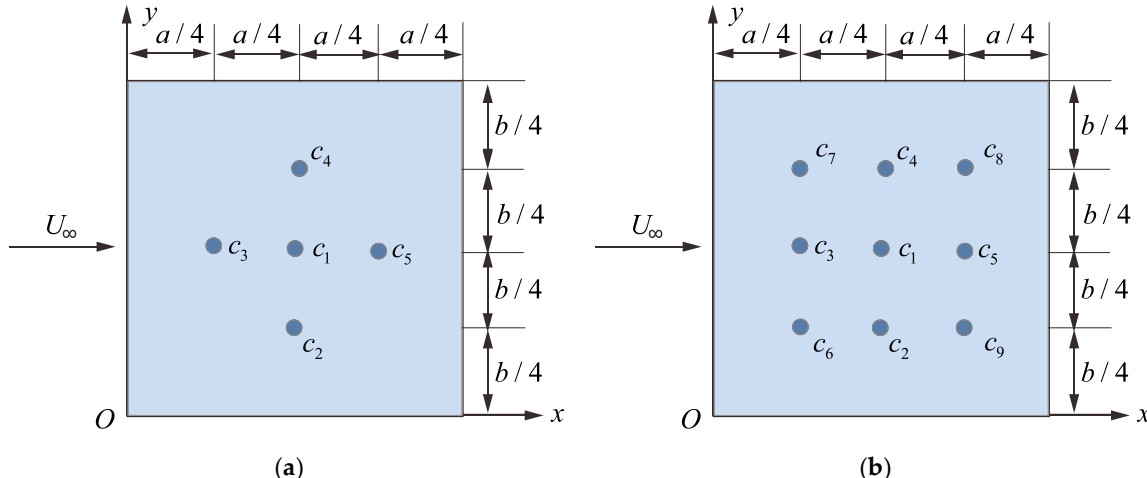

**Figure 15.** Two different damper configurations to suppress panel flutter with mass attachments: (**a**) 5P method, (**b**) 9P method.

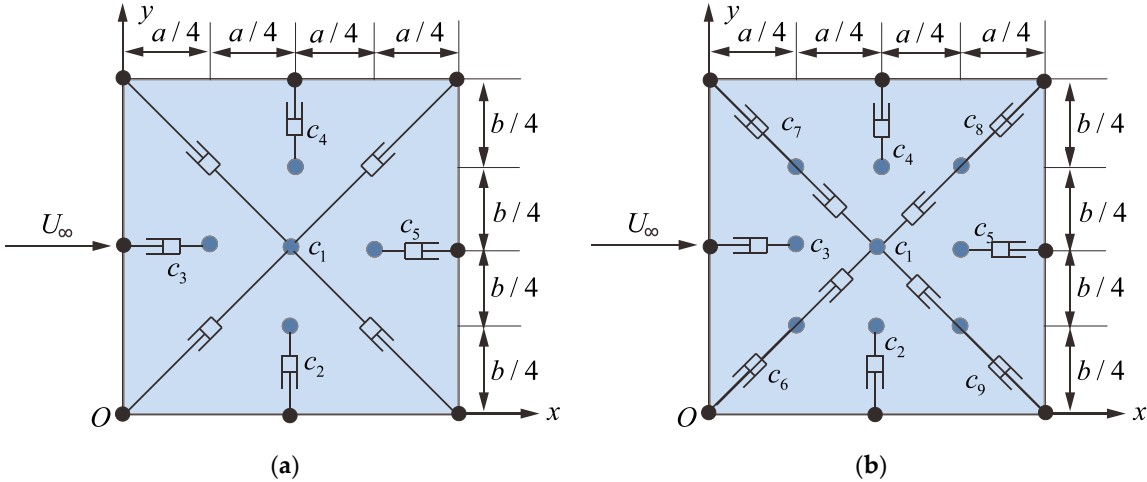

**Figure 16.** Two different damper configurations to suppress panel flutter with mass attachments: (**a**) 5P method, (**b**) 9P method.

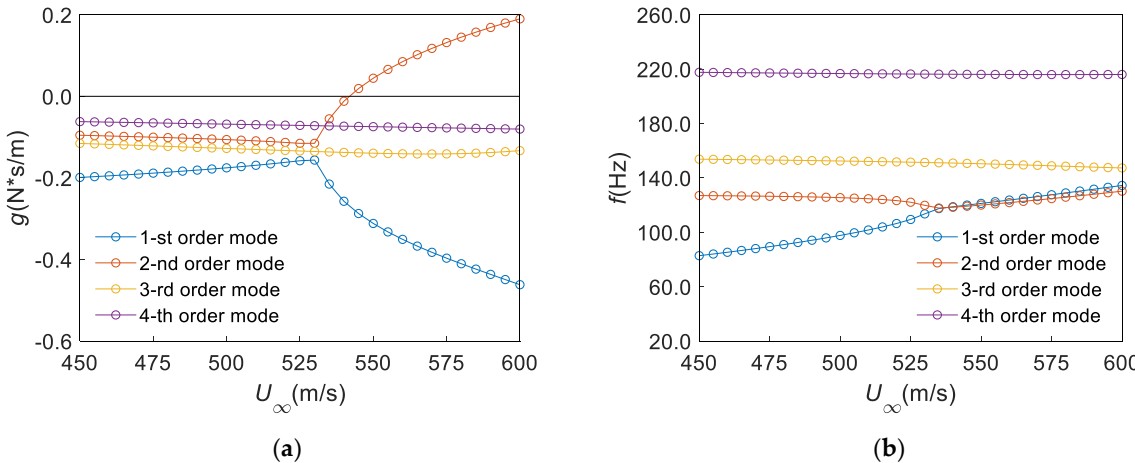

**Figure 17.** Results of flutter calculation for the panel with the 5P method damper configuration ($Ma_\infty = 2.0$): (**a**) $U - g$, (**b**) $U - f$.

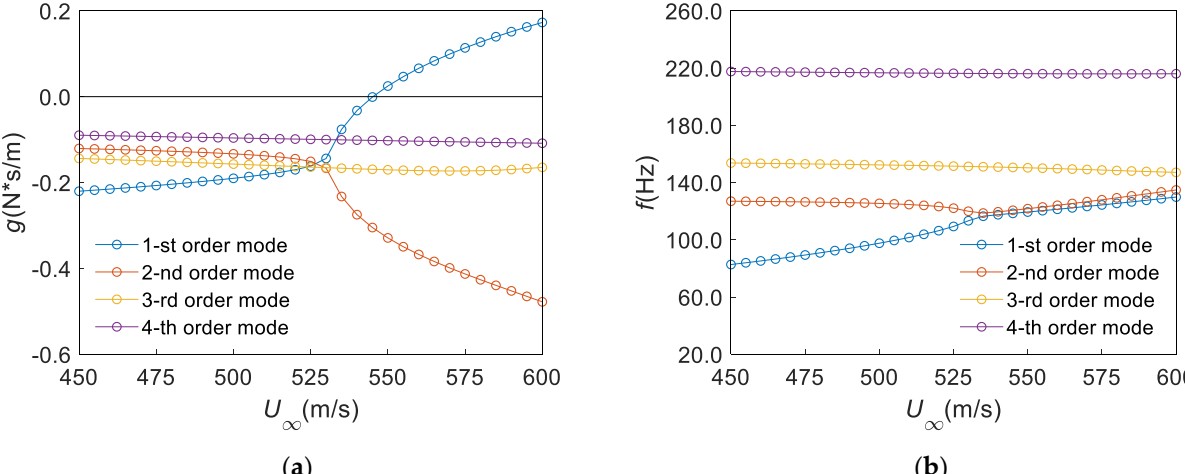

**Figure 18.** Results of flutter calculation for the panel with the 9P method damper configuration ($Ma_\infty = 2.0$): (**a**) $U - g$, (**b**) $U - f$.

Compared with the case of QP without damper, the flutter velocity of the panel based on the 5P method was increased by 5.6 m/s, with an increase rate of 1.04%. The flutter velocity of the panel based on the 9P method was increased by 8.9 m/s, with an increase rate of 1.66%. Similarly, the flutter frequency of the panel also increased. The flutter frequencies based on the 5P method and the 9P method were increased by 1.2 Hz and 1 Hz with an increase rate of 1.02% and 0.85%, respectively.

In order to better understand the gain brought by the configured dampers to the flutter characteristics of the panel, the case QP is still taken as an example. By changing the damping coefficients of all dampers from 0 to 10, we can observe the variation in the flutter velocities and frequencies of the panel as shown in Figure 19. It can be seen from Figure 19 that both the 5P method and the 9P method can effectively improve the flutter velocity and frequency of the panel. The flutter characteristics of the panel based on the 5P method basically increase linearly with the damping coefficient. However, the flutter velocities and flutter frequencies of the panel based on the 9P method increase faster than those based on the 5P method and show a trend of nonlinear growth. In particular, when the damping coefficient is 10, the flutter velocity and flutter frequency obtained based on the 9P method are 809.1 m/s and 180.9 Hz, respectively. Compared with the case of QP without dampers, the increasing ratios of the flutter velocity and the flutter frequency of the panel is 50.87% and 53.96%, respectively. Additionally, the 9P method, which suppresses the energy of the first four order main vibrations, has better enhancement effects on the flutter velocity and flutter frequency compared with the 5P method (only the first three orders are suppressed). The recovery and e enhancement of the flutter characteristics of the panel can be achieved by adjusting the damping coefficients of the dampers.

*6.4. Effects of Aspect Ratio on Flutter Characteristics of the Panel*

In this section, we discuss the effects of the aspect ratio on the flutter characteristics of the panel. For the convenience of comparison, the area of the plate remains unchanged. Two kinds of flat plates with different aspect ratios are shown in Figure 20. In Figure 20a, $a = 212.132$ mm and $b = 424.264$ mm, which meet $a/b = 0.5$. However, In Figure 20b, $a = 424.264$ mm and $b = 212.132$ mm, which meet $a/b = 2.0$. For convenience, we note that the cases shown in Figure 20a,b are the configurations of AR1 and AR2, respectively. Here, we only discuss the case where there is a mass attachment at point $(a/4, b/4)$, that is, the case of QP discussed in Section 6.1.

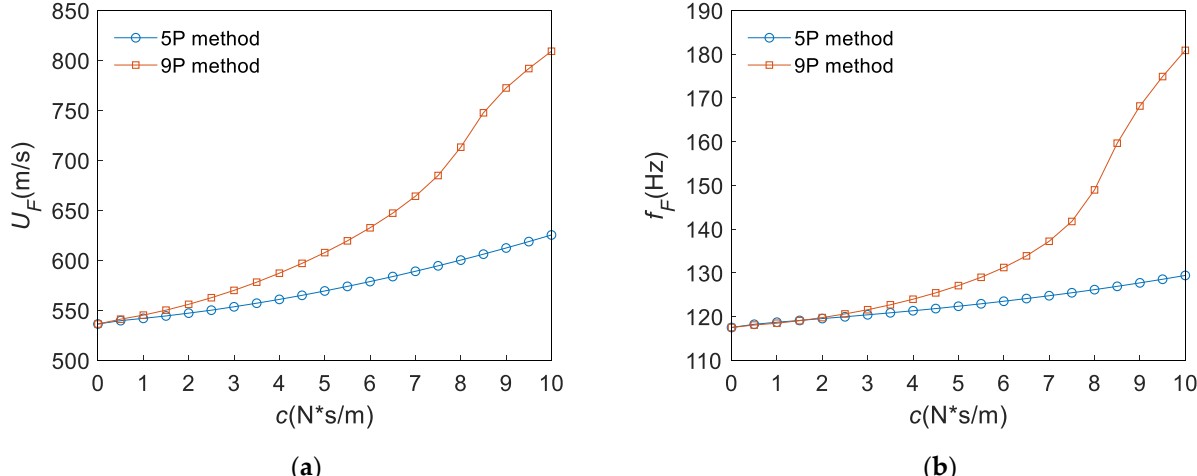

**Figure 19.** Effects of the two damper schemes on flutter characteristics of the panel: (**a**) $U_F - g$, (**b**) $f_F - c$.

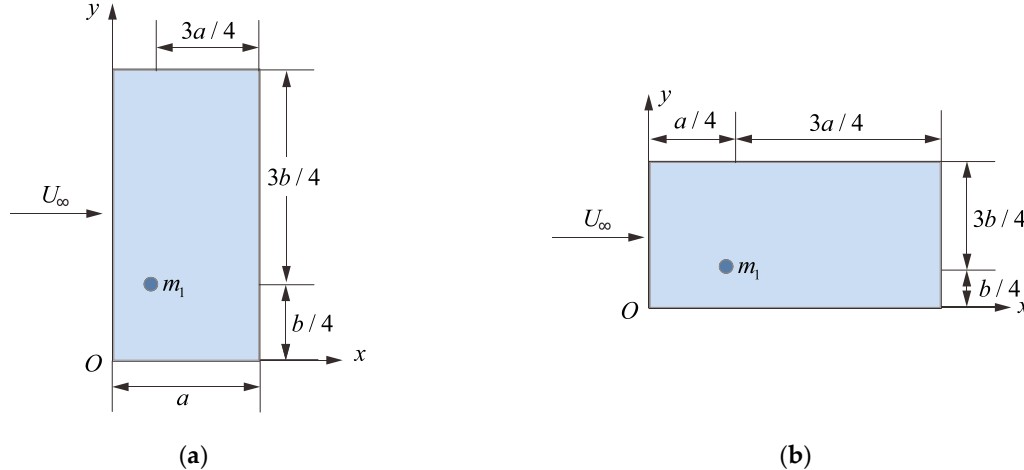

**Figure 20.** Two kinds of panels with different aspect ratio (QP): (**a**) $a/b = 0.5$ (AR1), (**b**) $a/b = 2.0$ (AR2).

First, we discuss the configuration of AR1. The results of the flutter calculation for the panel with no mass attachment are shown in Figure 21. It can be seen from Figure 21 that the flutter velocity and flutter velocity calculated by the program proposed in this paper are $U_F = 804.8$ m/s and $f_F = 230.0$ Hz, respectively. Flutter occurs in the fourth mode. It can be seen from Figure 4b that the first four modes are coupled together when flutter occurs. Then, the flutter calculation results under the case of QP are plotted in Figure 22. It can be seen from Figure 22a that the flutter velocity is $U_F = 774.9$ m/s, which decreases by 3.72% compared with the case of NP. At the same time, it can be seen from Figure 22b that the flutter frequency decreases by 5.87%. When dampers are configured on the wall panel using the 9P method (the damping coefficients of the dampers are all taken as 3.0), the flutter calculation results are shown in Figure 23. It can be seen from Figure 23 that the flutter velocity and flutter frequency of the panel with artificial damping configured according to the 9P method are $U_F = 806.3$ m/s and $f_F = 217.1$ Hz, respectively. Compared with the case QP, the flutter velocity was increased by 4.05%. However, the flutter frequency only increased by 0.28%.

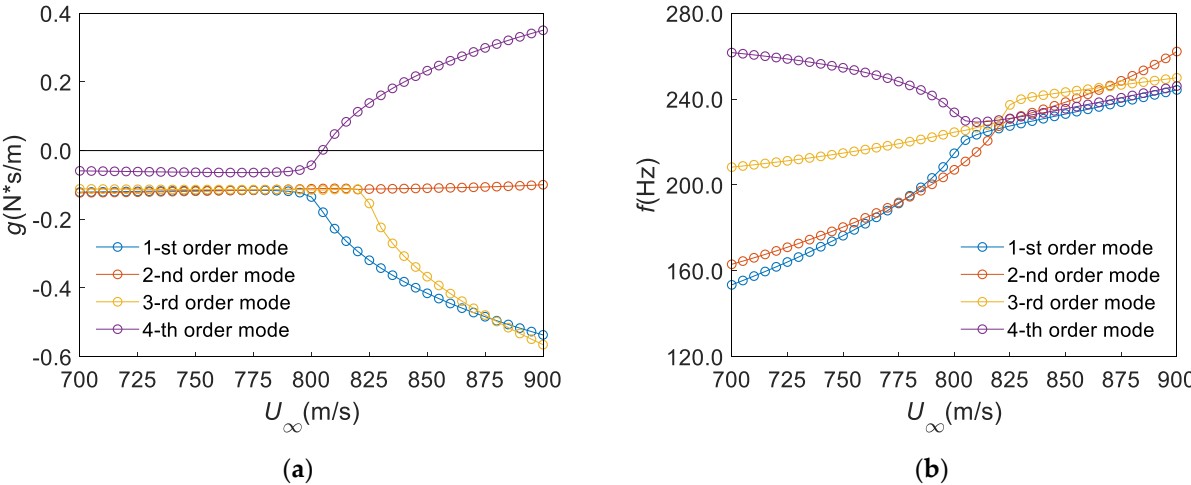

**Figure 21.** Results of flutter calculation for the panel with no mass attachment (NP) ($Ma_\infty = 2.0$): (**a**) $U - g$ (AR1), (**b**) $U - f$ (AR1).

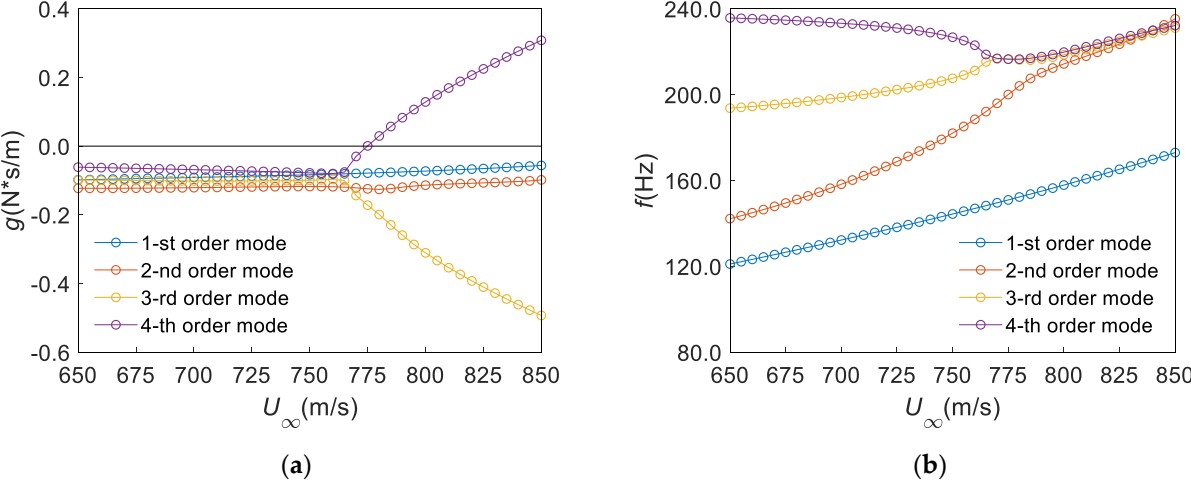

**Figure 22.** Results of flutter calculation for the panel with no mass attachment (QP) ($Ma_\infty = 2.0$): (**a**) $U - g$ (AR1), (**b**) $U - f$ (AR1).

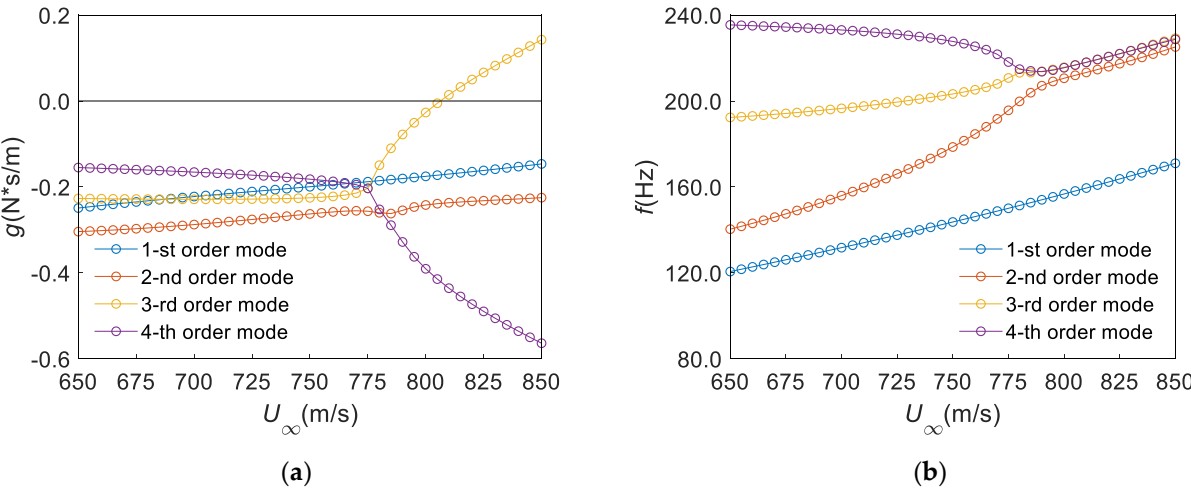

**Figure 23.** Results of flutter calculation for the panel with the 9P method damper configuration ($Ma_\infty = 2.0$): (**a**) $U - g$ (AR1), (**b**) $U - f$ (AR1).

Next, the configuration of AR2 is discussed. Figure 24 shows the results of the flutter calculation for the panel with no mass attachment. It can be seen from Figure 24 that the flutter velocity and flutter velocity are $U_F = 480.3$ m/s and $f_F = 113.0$ Hz, respectively. Flutter occurs in the first mode. It can be seen from Figure 4b that the first mode is coupled with the second mode when flutter occurs. Then, similar to the configuration of AR1, the flutter calculation results under the case of QP are plotted in Figure 25. It can be seen from Figure 25a that the flutter velocity is $U_F = 473.3$ m/s, which decreases by 1.46% compared with the case of NP. At the same time, it can be seen from Figure 25b that the flutter frequency decreases by 9.03%. Similarly, the damping coefficient of the damper is still taken as 3.0. The flutter calculation results are shown in Figure 26 when using the 9P method. It can be seen from Figure 26 that the flutter velocity and flutter frequency of the panel are $U_F = 546.8$ m/s and $f_F = 112.5$ Hz, respectively. Compared with the case of QP, the flutter velocity was increased by 15.53%. However, the flutter frequency only increased by 9.44%.

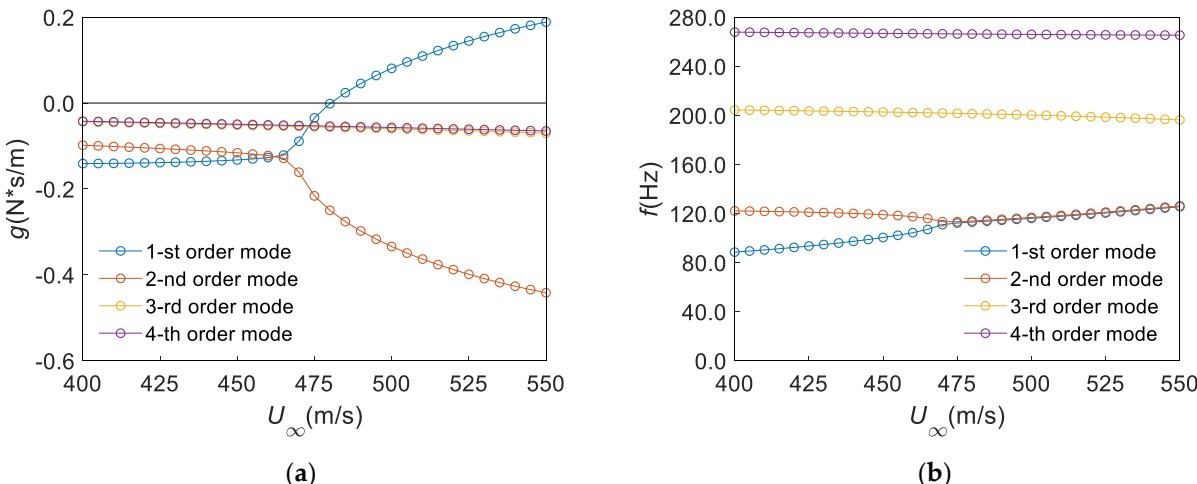

**Figure 24.** Results of flutter calculation for the panel with no mass attachment (NP) ($Ma_\infty = 2.0$): (**a**) $U - g$ (AR2), (**b**) $U - f$ (AR2).

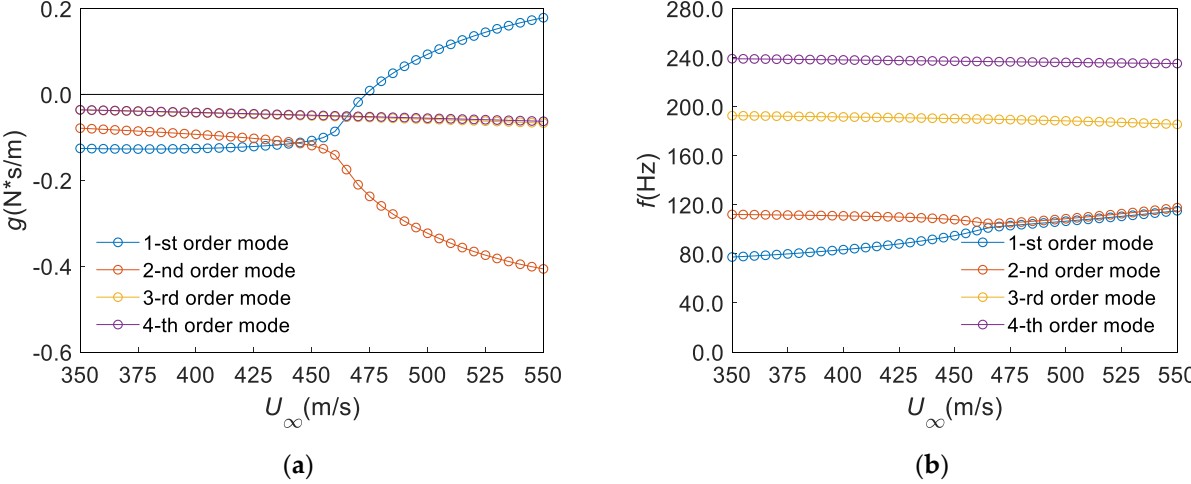

**Figure 25.** Results of flutter calculation for the panel with no mass attachment (QP) ($Ma_\infty = 2.0$): (**a**) $U - g$ (AR2), (**b**) $U - f$ (AR2).

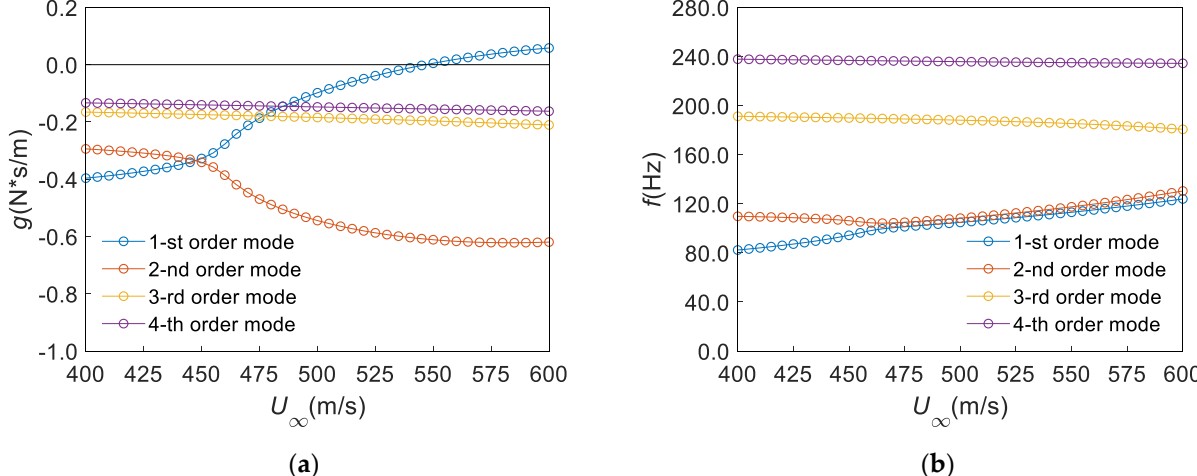

**Figure 26.** Results of flutter calculation for the panel with the 9P method damper configuration ($Ma_\infty = 2.0$): (**a**) $U - g$ (AR2), (**b**) $U - f$ (AR2).

Additionally, by comparing the flutter calculation results in Figures 21–26 with those in Figure 7, it can be seen that the flutter velocity and flutter frequency of the panel are higher than that of the square panel when $a < b$, while they are lower than those of the square panel when $a > b$. That is to say, reducing the length of the panel in the direction of air flow can effectively improve the flutter characteristics of the panel. In addition, we can draw conclusions that the flutter characteristics of panels with different aspect ratios change when they have mass attachments. Furthermore, the proposed damping scheme can enhance the flutter characteristics of panels with different aspect ratios and mass attachments.

## 7. Conclusions

Structural dynamic equations of panels with mass attachments were obtained based on the assumed mode method. Combined with the first order piston theory, the flutter calculation was carried out under different cases of mass attachments using the *p-k* method. Additionally, the corresponding flutter characteristic enhancement schemes by presetting dampers were provided. The main conclusions are as follows:

1. The proposed assumed mode method for panels with mass attachments can well capture the change in the natural frequencies of the panel structure. When there are mass attachments on the panel, the natural frequencies of the panel usually show a downward trend, but it may change dramatically due to different positions of mass attachments;

2. The changes in the flutter characteristics of the panel are closely related to the changes in the mass distribution of the panel caused by mass attachments. The flutter velocity of the panel can be improved by mass attachments located at the center of the panel. However, the flutter velocity of the panel drops sharply if the mass attachments are located in a nearby area slightly away from the central point. In addition, generally speaking, the flutter frequency of the panel with mass attachments is lower than that of the panel without a mass attachment. Furthermore, the larger the masses of attachments, the lower the flutter frequencies;

3. The flutter characteristics of the panel with mass attachments can be effectively improved by presetting dampers at appropriate locations. Both the present 5P method and the 9P method can effectively improve the flutter velocity and frequency of the panel, but the effect of the nine-point method is due to the five-point method. Additionally, the effects of the 9P method are better than that of the 5P method;

4. The study of the flutter characteristics of panels with mass attachments based on the assumed mode method can more realistically simulate the situation encountered

during flights. Additionally, it can provide technical reserves for the structural design of panels and the improvement of flight safety.

**Author Contributions:** Conceptualization, W.Q. and S.T.; methodology, W.Q.; software, M.W.; validation, W.Q.; formal analysis, W.Q.; investigation, M.W.; resources, S.T.; data curation, M.W.; writing—original draft preparation, W.Q.; writing—review and editing, S.T.; visualization, M.W.; supervision, W.Q.; project administration, W.Q.; funding acquisition, W.Q. All authors have read and agreed to the published version of the manuscript.

**Funding:** This research was funded by the National Natural Science Foundation of China, grant number 11502149 and 11902204, and the Project of Liaoning Provincial Department of Education, grant number JYT2020034 and JYT2020029.

**Institutional Review Board Statement:** Not applicable.

**Informed Consent Statement:** Not applicable.

**Data Availability Statement:** Not applicable.

**Conflicts of Interest:** The authors declare no conflict of interest.

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
