# Peer review of "Effects of Mass Attachments on Flutter Characteristics of Thin-Walled Panels"

_aerospace, doi:10.3390/aerospace9120748_

Round 1

Reviewer 1 Report

In this work, the effect of concentrated mass on supersonic panel flutter is investigated. To my knowledge, such a problem formulation was not studied before, and the results of this paper are interesting for aeroelasticity. I recommend publication after minor revision according to the following comments.

1. The authors consider 16 (4x4) modes to represent modes of the panel with mass attachments through modes of the panel without attachments (line 285). However, there is no evidence that this number of modes is sufficient. Please discuss your convergence study.

2. Flutter in Fig. 5 and similar figures occurs due to coupling of the first and the third modes. I believe those are (1,1) and (2,1) modes, where the first and the second numbers denote number of half-waves in the streamwise and spanwise directions. However, the (2,1) mode is commonly called "second" (not the "third") mode. Moreover, looking at fig. 5, it is clear that at small speed, (1,2) and (2,1) mode frequencies coincide, i.e. calling one of them "second" and the other "third" is an arbitrary choice. However, with the flow speed increasing, it becomes clear that (2,1) mode frequency becomes smaller that (1,2) frequency, that is why it is indeed natural to call (2,1) mode the "second" mode. After that, the flutter will take place after the coupling of the first and the second mode, as is usual in literature. I recommend to revise the manuscript and renumber the modes accordingly.

3. Finally, a note on the piston theory limitation can be made. Namely, single mode flutter occurring at M<2, cannot be detected by the use of piston theory. This type of flutter is not considered in this work. The papers of Abdukhakimov et al. (AIAA J, 2022) and Shishaeva et al. (AIAA J, 2018) can be discussed in this regard.

Reviewer 2 Report

1. The authors can elaborate by examples on the various mass attachments found on an aircraft panel in practice (other than ice being one of them as mentioned by the authors)

2.The authors can mention the practical ways of application of damping approaches used for damping of panel vibrations in Aircraft design.

3.The authors have presently assumed a square panel in their examples. Would the conclusions arrived be valid for panels of other shape e.g: rectangular? Please discuss.

4. Figure 7 (c) can be re-plotted with proper displacement scaling as it stands out in comparison to other associated figures in Figure 7.
